**Data Availability Statement:** All relevant data are within the manuscript and its Supporting Information files.

# Human scent signature on cartridge case survives gun being fired: A preliminary study on a potential of scent residues as an identification tool

**Nikola Ladislavová**[1☯]*, **Petra Pojmanová**[1☯], **Pavel Vrbka**[2☯], **Jana Šnupárková**[3‡], **Štěpán Urban**[1‡]

**1** Faculty of Chemical Engineering, Department of Analytical Chemistry, UCT Prague, Prague, Czech Republic, **2** Faculty of Chemical Engineering, Department of Physical Chemistry, UCT Prague, Prague, Czech Republic, **3** Faculty of Chemical Engineering, Department of Mathematics, UCT Prague, Prague, Czech Republic

☯ These authors contributed equally to this work.
‡ JS and SU also contributed equally to this work.
* ladislan@vscht.cz

## Abstract

This paper focuses on a chemical analysis of human scent samples that were obtained from cartridge cases after being fired and their comparison with scent samples collected under laboratory conditions. Scent samples were analyzed by comprehensive two-dimensional gas chromatography coupled with the time-of-flight mass spectrometer. The results obtained from the chemical analyzes confirmed the desired stability of the human scent evidence and outlined the possible application for forensic purposes. The qualitative results of the study converge with the findings of previous studies on the composition of human scent and the chemical composition of human fingerprints. Furthermore, statistical analyzes were performed employing similarity algorithms such as Pearson's and Spearman's correlations, or Kendall's tau. The resulting comparison of the scent samples secured on fired cartridge cases compared with those samples collected under laboratory conditions yielded ten out of ten correct identifications of the scent inflictor.

## Introduction

Nowadays, standard forensic procedures regarding firearms and crime scenes involve finding and securing fingerprints, gunshot residues, and scent traces. Specifically, human scent traces, if secured from a gun handle or gun slide on the crime scene, are generally compared by specially trained dogs [1]. The ability of trained dogs to distinguish between human individuals based on their 'scent signatures' is well known and described [2–4]. Human scent is a complex mixture of thousands of chemical compounds, mainly hydrocarbons, heterocompounds such as ketones, sterols, sterones, fatty acids, and esters of fatty acids [5–8]. Although the chemical character of the scent compounds could suggest a lesser resistance of the scent traces when

**Funding:** Ministry of Interior VJ01010123 Ministry of the Interior of the Czech Republic https://starfos. tacr.cz/cs/project/VJ01010123 The funders had no role in study design, data collection and analysis, decision to publish, or preparation of the manuscript.

**Competing interests:** The authors have declared that no competing interests exist.

exposed to heating, fire, and explosions, there are studies indicating that the scent signatures survive such conditions [9, 10]. This phenomenon can be explained by the multiplicity of scent signature. The study of Doležal et al. [11] confirmed that specially trained dogs can identify individuals by different chemical fractions of their scents. In addition, the least volatile fraction of chemical compounds proved to be the most relevant for olfactory identification. However, tests performed by specially trained dogs are very often challenged in court as there is no concrete numerical proof of the results. The aim of this study was to acquire evidence whenever the human scent can survive extreme conditions during the gunshot, to analyze the scent residues by two-dimensional chromatography coupled with mass spectrometry and perform comparison analyzes while mimicking the procedure performed by specially trained dogs. The results indeed supported the assumption that it is possible to secure human scent traces from post-blast items (fired cartridges) and compare such residue samples against samples collected under the laboratory condition.

## Materials and methods

### Ethics statements

This study was approved by the Institutional Review Board of the University of Chemistry and Technology, Prague (approval number EK/8/2020) and complies with the Declaration of Helsinki for Medical Research involving Human Subjects.

Volunteers gave their written consent to collect, analyze, and store their scent samples (S2 File)–for this subbranch of research of human scent, volunteers were not asked to fill questionary data, as those data were outside the scope of the experiment.

### Volunteers

In total, twenty volunteers (ten males and ten females aged from 25 to 40 years) provided their scent samples through the experiments. Repetitive scent collection took place approximately once a week at the same place at the same time.

### Sampling procedure and sample collection

First, the volunteers washed their hands with non-perfumed soap (Cormen, CZ), rinsed them with a warm tap water until there were no soap bubbles, and let the hands dry in the air to complete dryness. Subsequently, the volunteers rubbed their palms to activate the scent glands. After 5 minutes, the volunteers received cartridges (S&B 7.65 Browning). The sampling itself continued for 10 minutes. Volunteers were holding and rubbing the projectiles in both hands. Once the collection procedure was over, the cartridges were transferred to vials and sealed. Comparative samples on glass beads were collected with the same procedure and were referred as standard scent samples in the following text. In an experimental shooting room, the cartridges were loaded by a designated person (a researcher who was not involved in the volunteer sample pool) into the pistol and fired (the shooter was another researcher who was not involved in the volunteer sample pool). The fired cartridges were left to cool down on the sterile surface (waterproof surgical drapes) and then moved into the vials (see Fig 1). Both persons present at the shooting site were wearing surgical gloves and respirators to prevent sample contamination.

The time between both sampling procedures (glass beads sampling and projectile sampling) was as short as possible (less than 12 hours). The extraction procedure of the collected scent samples was the same for both the scent carriers, for the glass bead samples and for the fired cartridge cases samples. First, 2 mL of ethanol ($> = 99.8\%$, Penta CZ) were added to each vial with the collected scent sorbent. At this initial stage of extraction, the sample vials were placed

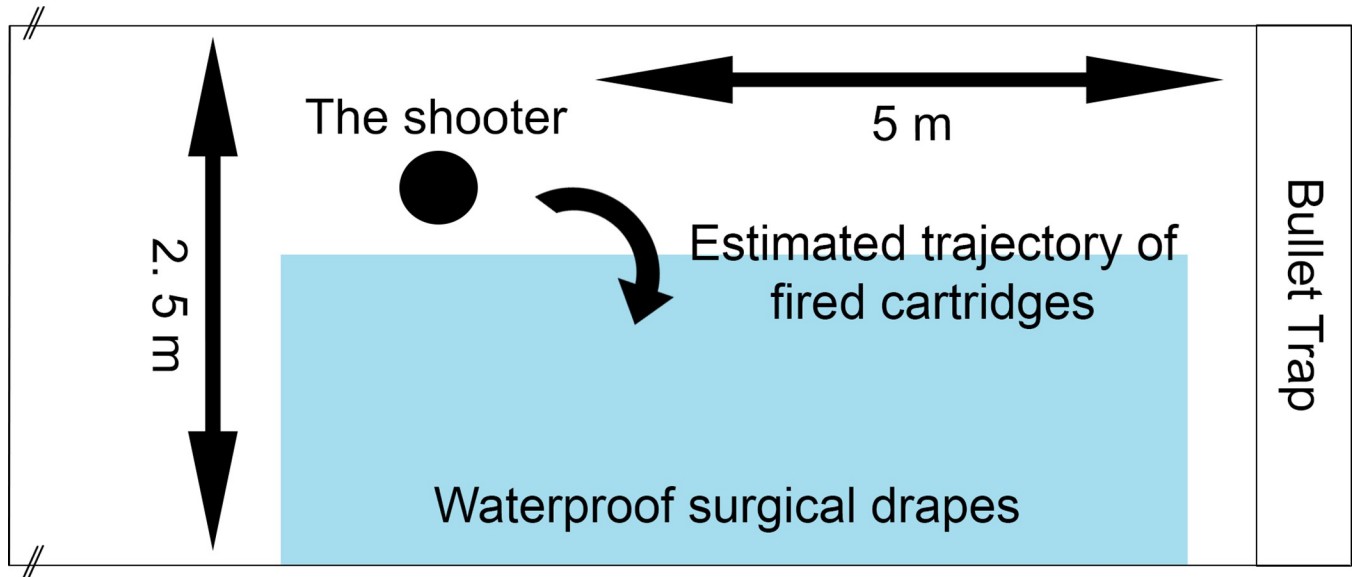

**Fig 1. The sketch of experimental shooting range.** The waterproof surgical drapes were placed on the floor to the area of estimated landing spot of the fired cartridges.

in an ultrasonic bath (Fisher Scientific, UK) for 10 minutes and then on a shaker (1000 rpm, IKAVIBRAX VXR basic, Germany) for an additional 10 minutes. The solution was pipetted into 2 mL champagne vials and dried to complete dryness under reduced pressure (concentrator Genevac EZ-2, USA). Right before the analysis, the sample was dissolved again in 70 μL of ethanol. For contaminant identification purposes, blank samples of glass beads and fired cartridge cases were taken for each sampling procedure.

## Methods

All analyzes were performed by the GC×GC system with a LECO Pegasus 4D-C detector (LECO Corp, USA), a 7890B gas chromatograph (Agilent, USA), and a MPS robotic MPS multipurpose sampler autosampler from Gerstel (Germany). The first column was a 30 m long Rtx-200 column (0.25 mm; 0.25 μm—Restek, USA) with an additional 2 m as the precolumn. The second column was a 1 m Thermo-5HT (0.25 mm; 0.25 μm—Thermo Scientific, USA). Helium (purity $\geq$ 99.9995 from Linde CZ) was used as the carrier gas with the flow set at 1.5 mL / min. The temperature program was the following: 40˚C—2 minutes hold—5˚C/min ramp—320˚C—10 minutes hold. The modulator offset against the secondary oven was set at 15˚ C and the secondary oven offset against the primary oven was set at 5˚ C. There were three modulation periods during the run: 6s (start→1700[th] s), 8s (1700[th] to 2588[th] s), and 10s (2588[th]→ end). The injection chamber was heated to 280˚ C. The injection volume of the sample was 1μL (splitless). The temperature for the transfer line was 280˚ C and for the ion source was 250˚ C. The electron ionization energy was 70eV. 29–800 m/z fragments were scanned at rate of 200 spectra per second. This method was previously evaluated as the best possible option for measuring human scent samples [12].

### Data processing

The LECO® software (version 4.72.0.0, LECO Corp., USA) was used for the basic evaluation of the measured chromatograms. The retention indexes (RI) were calculated for each

chromatogram to treat the time shift in the first dimension. The RIs could not be based on the elution of *n*-alkanes because of their lack or overlap with the contaminant in the collected samples. All RI calculations were based on siloxanes. The main advantage of the siloxane approach was their presence as contaminants, even in the blank samples, which was crucial for the automatic sorting of the detected peaks. After data alignment via RIs, all detected peaks were inspected, and the subset of 75 selected chemical compounds presented in standard and fired sample datasets was created and further processed. Each sample, its peaks, and their areas, respectively, were transformed to a matrix of peak area ratio (*75×75* dimension–if the chemical compound was absent in all inspected samples, the dimensionality was reduced) and the matrix was then rewritten as a single row representation. Peak area ratios where the divisor was equal to zero were set to zero. The peak area ratios with the 'infinite' value were also approximated by zero. Eliminating the peak area ratios with zero values was the next step in the data processing procedure. The selection algorithm for the 'significant' peak area ratios for model-based analyses was the following: For each volunteer, only the peak area ratios with incidence rate ($T_1$) of 75% (for example, peak area ratios had a nonzero value in eight out of ten samples) were chosen. It means that the *zero_count* parameter in the used script was set to 25% (incidence thresholds $T_x$ are equal to 100% − *zero_count* parameter). Then, the *N* number of peak area ratios with the least standard intra-subject deviation (sorted in ascending order) through all samples were included in further calculations. The *N* parameter was estimated and optimised based on the results of the PCA. The *N* parameter can range from 1 to all the peak area ratios. The *N* range tested in this study ranged from 25 up to 500 with a step of 25). Finally, the 'significant' peak area ratios of all volunteers were merged and, again, the incidence rate threshold ($T_2$) was established. The $T_2$ was also estimated and optimised in range from 25% up to 75% with step of 25%. When only the most common peak area ratios were involved in all samples, the $T_2$ threshold was higher. If more '*volunteer-specific*' peak area ratios were targeted in the data set, the $T_2$ threshold was set to lower values. The models are named after the used parameters: Model_*Nparamater_T2parameter*.

## Data comparison and evaluation

The normality of the experimental data set was tested with the multivariate normality test, specifically Mardia's test [13]. Information about the (non)normality of the distribution was important for the correct estimation of the parameter *N* value and an evaluation of the results from the similarity comparison. There were two main strategies for setting the *N* parameter. First one was to set rigid standard deviation threshold value, but this approach was not suitable for non-normal distributed data. Therefore, an approach of setting the rigid count of the peak area ratios with the least standard deviation was prioritized. Two different statistical approaches were used for the following data analyzes: Principal Component Analysis and similarity comparison method. Principal Component Analysis (PCA) [14] was used solely as a visualization of possible trends in data variance and as a tool for parameter optimization–models with higher percentage of variance explained for 7 principal components were considered as more reliable. Different sets of zero-value filters (incidence rate threshold) and number of characteristic ratios (*N*–see section Data processing) were tested. Subsequently, the best PCA models (models that describe the most variance in dataset and with separated volunteer clusters) served as the basis for similarity analyses in the next stages of our experiment. In addition, K- Nearest Neighbor (KNN) [15], Radius Nearest Neighbor [16] (RNN), and Random Forest (RF) [17] classification algorithms were applied on samples used in the model creation steps. All three algorithms were optimized via Grid Search algorithm [18] with 5-fold cross validation with *accuracy* as the scoring argument, and the data had a train/test split ratio of 75% /

25%. The KNN was optimized on grid: *K* in range from 1 to 19 with *weights* set to *uniform* or *distance weighted*. The RNN had same weights grid setup and the *radius* parameter was tested in range from 1 to 99 with step of 10. The RF classification model was optimized on the following parameter grid: *number of estimators*– 1 to 200 wit step of 10; max_*features*– 1 to 11 with step of 2; *bootstrapping* set to active or inactive, and if the *bootstrapping* was active, the *oob scoring* was tested on active and inactive level as well. For complete analysis of the data, first three principal components of PCA models were feed to the machine learning models (KNN, RNN and RF) and compared with the performance of models applied on original data. $F_1$ [19] was used as the performance metric. Regarding similarity comparison, four similarity algorithms were used: Cosine similarity (CS) [20], Pearson's correlation (PC) [21], Spearman's correlation (SC) [22, 23], and Kendall's Tau correlation (KT) [24]. Mardia's test was performed as a built-in function of the Unscrambler® software (version 10.5). The PCA, KNN, RNN, RF, CS, PC, SC, and KT scripts were written in Python using Matplotlib [25], Numpy [26], Pandas [27], Scikit-learn [28] and Scipy modules. All written scripts are published in the opened repository (see S1 File).

Fig 2 depicts a workflow of data processing and data comparison steps from the creation of similarity models to the final application of best performing models on standard scent samples versus shot cartridge samples comparison.

## Results and discussion

### Projectiles and fired cartridges as sampling materials

First, a test was performed whenever the surface of the projectile is usable as a scent sampling material. The scent samples collected on the projectiles were qualitatively similar to those collected on the glass beads. However, the chromatograms of non-fired projectiles were, in general, less intensive than those of the glass beads' sample. This means that the number of detected peaks and areas of detected peaks were lower than in the samples collected on glass beads. This could be the result of the surface characteristics (relief) of both sampling materials, as both glass and metal are, from the human scent point of view, inert to the collected samples. After it was proven that metal cases of projectiles can be used as a sampling material for sample collection, blank chromatogram and cartridge scent sample chromatogram (the metal case of the projectile left on the floor after shooting) were measured. Chromatograms of standard scent sample collected on glass beads, blank projectile, blank fired cartridge, and fired cartridge with sampled scent chromatograms are demonstrated in the (S1 Fig). The chromatograms of the blank projectiles and blank fired cartridges were qualitatively analogous to blank samples of glass beads. Furthermore, the blank samples of the fired cartridges contained a 'cloud' of aliphatic, both branched and non-branched alkanes, alkenes, and cyclic compounds (S1 Fig, areas A and E)–this contamination likely comes from lubricating oils used for the maintenance of the weapons.

### Scent traces on fired cartridges

The overall chemical composition of the scent trace endured conditions during the weapon's firing. However, the elution area of fatty acid esters and less volatile compounds (S1 Fig, area C) contained a smaller number of peaks, and the peaks were generally less intensive (as compared to standard scent samples). The elution pattern of the most frequent chemical compounds (such as ethyl esters, amines, and steroid derivates) and their summary are shown in S1 Fig and S1 Table. Considering the nature of the compounds found (mostly $C_{16}$ and higher), it was clear that the volatility of the chemical compound would be a key factor in the study. The higher ethyl esters were previously reported in human scent samples [29, 30]. Squalene

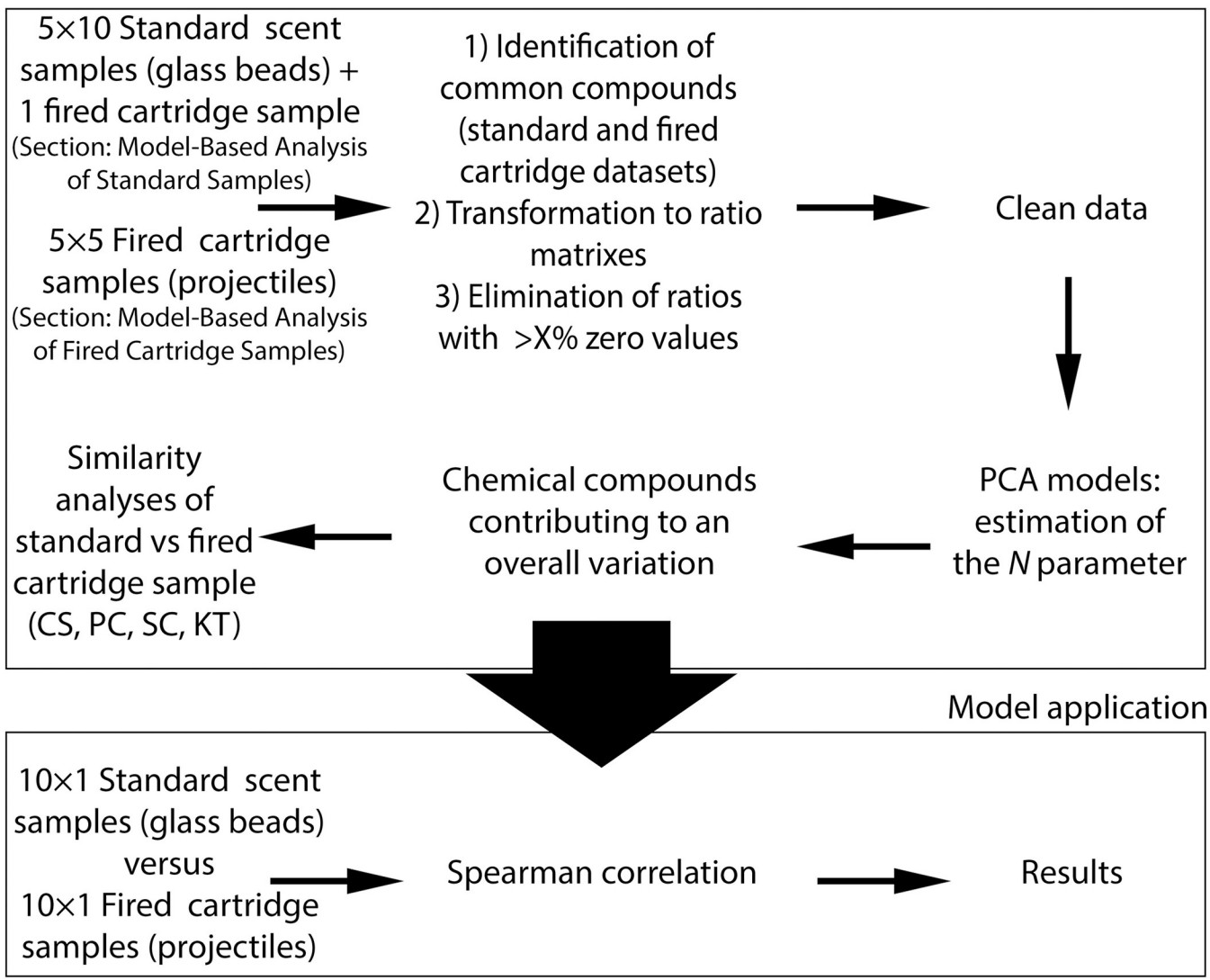

**Fig 2. Scheme of an experimental process.** For the model development (top box), two different datasets were used. Five volunteers sampled their scent on glass beads for ten times and additionally, they sampled their scent on a projectile once as well. Another five volunteers sampled their scent on the projectiles five times (in total, four samples were excluded from further calculations as they exhibited severe contamination). The rest of the top box describes general steps for the selection of the best performing models. The bottom box depicts the final application of the model on the data set of ten volunteers who gave one sample on glass beads and one sample on a projectile each.

[30, 31] was also found in the cartridge samples, but its´ elution area overlapped with the "aliphatic cloud". Thus, it was excluded from the further calculations. Nitrogen compounds as derivates of urea or nitrogen heterocycles were also reported in samples of human scent [29]. Steroid chemical compounds have also been reported [32]. In addition, steroids were marked as a 'scent precursors' [7, 33] and products of the sebaceous glands along with squalene, fatty acids and their esters [34]. Furthermore, compounds of the same chemical species were detected by gas chromatography in the samples of a fingerprints [35]. Note that only a mass spectra comparison was done for the identification of the detected chemical compounds. Thus, the findings are discussed as a chemical group, not as the specific chemical compounds.

## Model-based analysis of fired cartridge samples

First, it was necessary to determine whether the remaining scent compounds on the fired cartridges included enough information for the distinction between the volunteers. The PCA model was built on ratios with the least standard deviation (SD) through the samples of each volunteer. The best model (Fig 3) was based on $N = 75$ ratios. However, to reduce the peak noise and keep the relevant values, we considered ratios with more than 50% nonzero values ($T_2 = 50$%) through all samples (Model_75_50) only. The experimental data set exhibited a non-normal distribution (normal skewness, but kurtosis differed significantly). Originally, five volunteers provided five samples for this experiment, but one volunteers´ set exhibited severe contamination and therefore it excluded from the dataset. Additionally, two samples from two volunteers (Vol1 and Vol4) had to be excluded due to the severe contamination as well.

KNN and RF trained models had $F_1$ score equal to zero. The RNN model failed the classification and the $F_1$ score was equal to 0.5.

**Similarity analysis of fired cartridge samples.** One random sample from each volunteer was chosen as the 'unknown' sample, and the rest of the samples ('standards') were compared to that 'unknown' sample. Every classification step was successful with the four approaches (CS, PC, SC, and KT) (see Fig 4).

## Model-based analysis of standard samples

Five volunteers produced 11×5 samples in total, 10 standard scent samples, and one projectile sample. However, three standard samples ($2 \times$ Vol 2 and $1 \times$ Vol 5) were excluded from further

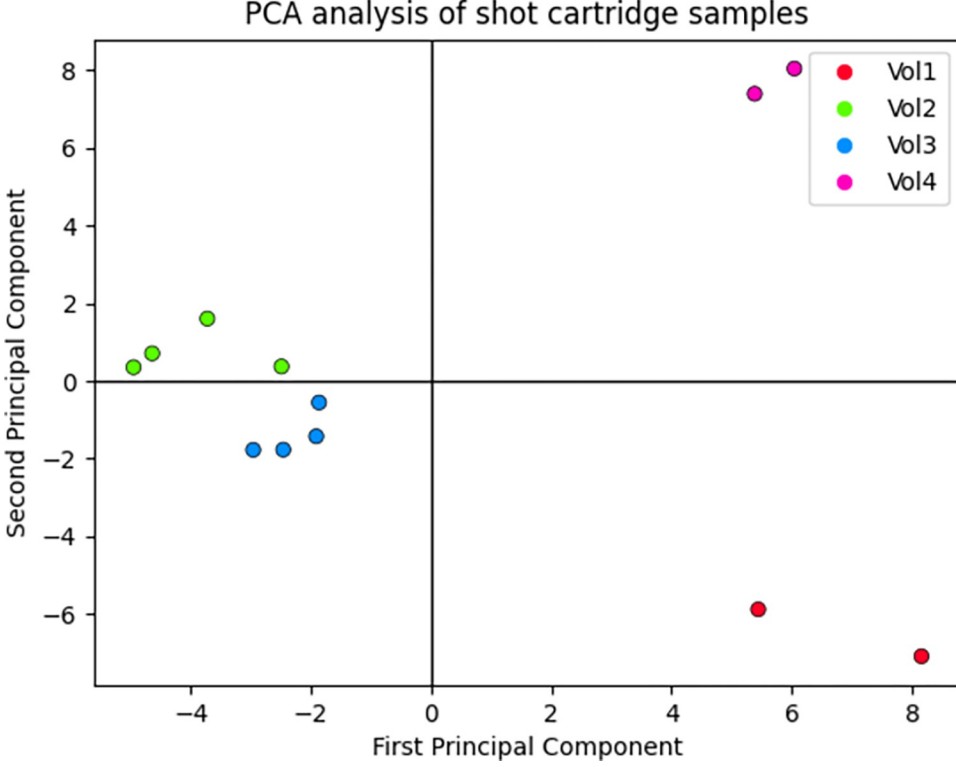

**Fig 3. PCA of scent samples collected from shot cartridges.** This model was based on $N = 75$ ratios with the lowest SD for each volunteer. The $T_2$ was set to 50% non-zero values for each ratio across all samples. 65 peak area ratios are the intersection of all volunteer sets, and these ratios were present in more than 50% across all inspected samples. The first two components described 59.75% of the data variance. Further comments on the PCA analysis setup are stated in S1 File.

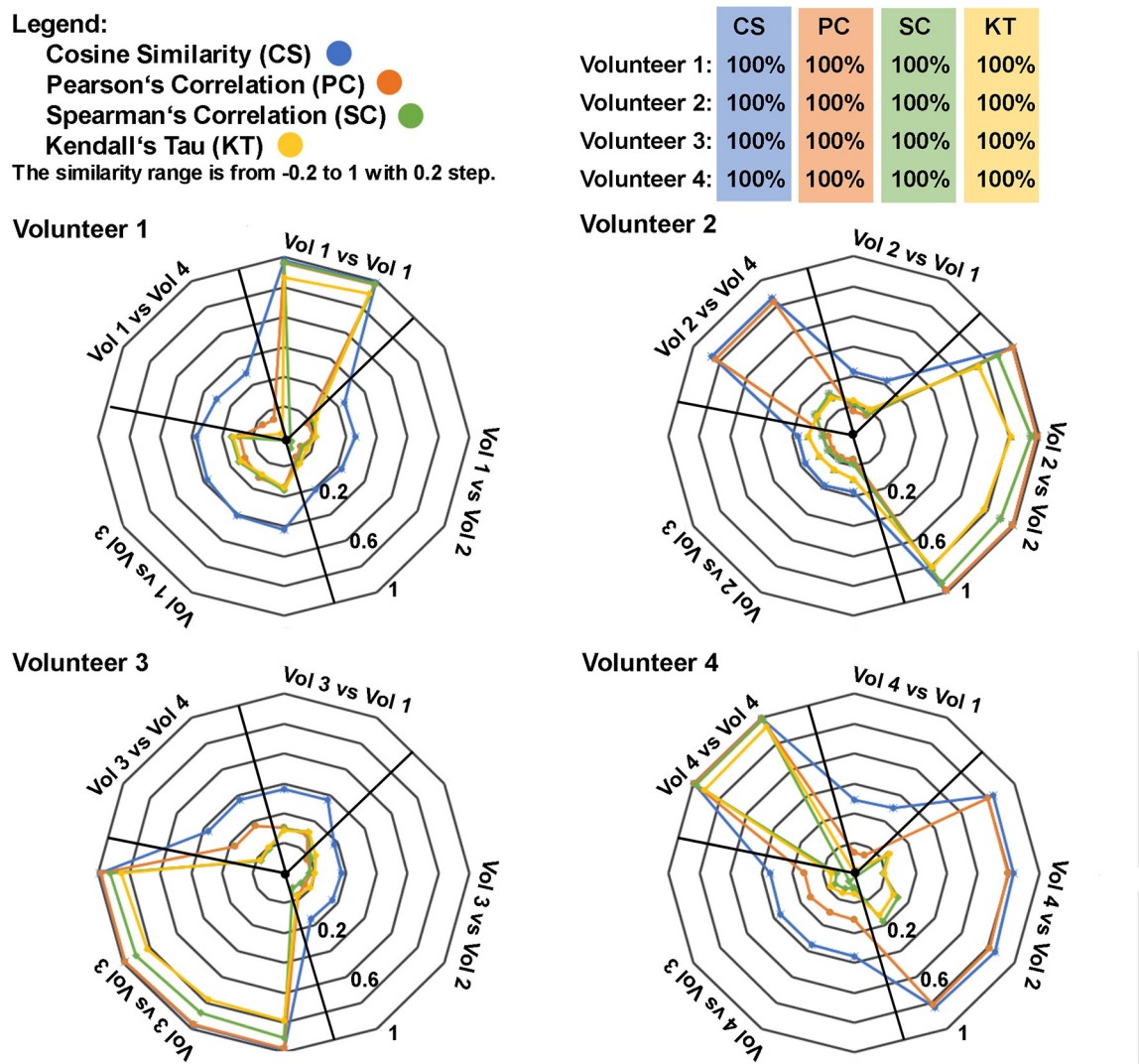

**Fig 4. Results of similarity between randomly selected fired cartridge samples (examined) from each volunteer compared to the rest of the fired cartridge samples.** There were two referential samples from Vol 1, four from Vol 2, four from Vol 3, and two from Vol 4. The closer the line is to the edge of the circle, the more similar the compared samples.

analyses due to low intensities (due to sampling or handling error, the measured chromatograms were almost blank). The initial PCA model was constructed on the ratios that were considered in Model 75_50 (Section Model-based analysis of fired cartridge samples). PCA analysis of standard samples based on this model failed to reveal differences between volunteers. Another model was built, and multiple least SD ratio setups were tested. The best discrimination power was achieved with $N = 50$ ratios (for each volunteer) and with the $T_2$ set to 25% across all inspected samples as demonstrated in Fig 5. The experimental data set exhibited a non-normal distribution (skewness and kurtosis differed significantly from the normal distribution).

The best performing KNN model reached $F_1$ score equal to 0.85. RNN model best $F_1$ score was equal to 0.60. The RF performed with the $F_1$ score equal to 1. The confusion matrix of the RF model is shown in the Fig 6. The results and the model parameters are supplied in the (S1 File). For the completion of the data inspection, all models were retrained on the reduced data

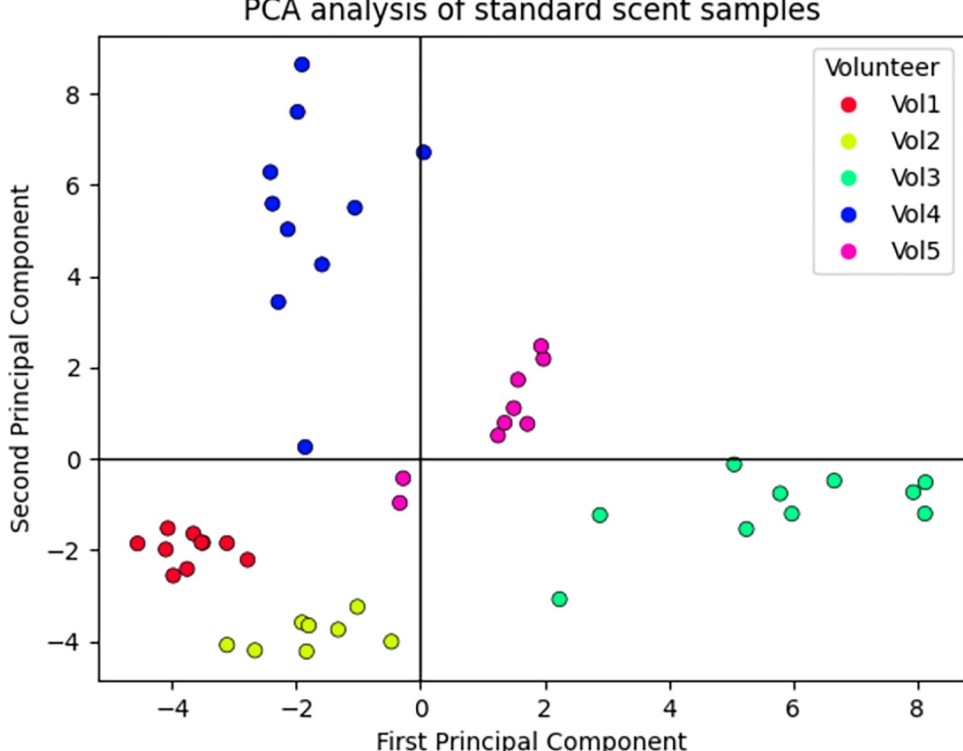

**Fig 5. PCA of the scent collected from glass beads samples and a shot cartridge sample.** This model was based on $N$ = 50 ratios with the least intrasubject SD for each volunteer. The $T_2$ was set to 25%. 57 peak area ratios are the intersection of all volunteer sets, and these ratios were present in more than 25% of the samples. The first two components described 42.10% of the variance in the data. Further comments on the PCA analysis setup are given in S1 File.

(first three principal components). RF model retained $F_1$ score equal to 1. $F_1$ score improved for both NN models: KNN scored 0.92 and RNN scored 0.85 in terms of $F_1$ score. The similarity analyses were carried out on the non-reduced (original) data, thus, the performance of the models when applied on original data was more significant for the *proof-of-concept* tests.

## Similarity analysis of fired cartridge samples versus standard scent samples

The Spearman's correlation coefficient was the best performing correlation from all tested approaches (for the full representation of the results see Fig 7 and S3 Table). For Volunteers 1, 3, and 5 all their standard scent samples had higher correlation coefficient compared to the fired cartridge sample than the standard samples of the other volunteers. For Volunteer 2, one standard sample a had lower correlation coefficient compared to the fired cartridge sample than the standard samples of another Volunteer. The same case occurred for Volunteer 4. Cosine similarity was achieved at a level similar to that of SC. It was correct in all cases for Volunteers 1, 4, and 5, and made one error for Volunteers 2 and 3. Kendall's tau reached the top score for the same volunteers as SC did, and similarly, made one error in the case of Volunteer 2. However, KT made two errors when comparing Volunteer 4's samples. Pearson's correlation reached the highest score only in two cases, Volunteer 1 and 5. PC made one error in comparing the samples of Volunteer 3 and 4, and two errors for Volunteer 2. The reason why SC and KT comparisons achieved better results is probably that, unlike PC, these correlations are more suitable for data with a non-normal distribution.

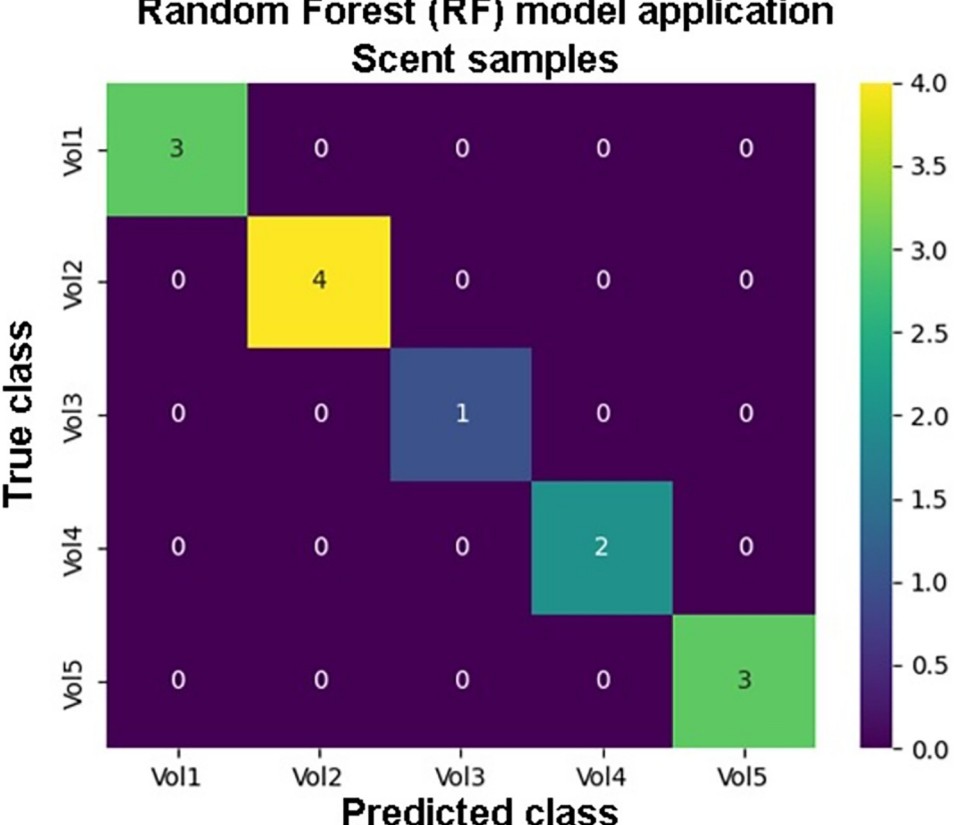

**Fig 6. Confusion matrix for the best performing RF model.** The train/test split ratio for the model training was 75%/25%.

## Fired cartridge sample versus standard scent sample comparison: Model application

For the final comparison, a group of ten volunteers different from the group from previous experiments was asked to provide two samples–one collected on the glass beads ('standard sample'), and one collected on the projectile ('fired cartridge sample'). The samples were treated as stated in section Sampling procedure and sample collection. The structure of an experiment was designed to simulate the same procedure as that performed by specially trained dogs [2]. Each shot cartridge sample was compared to ten standard samples–a *comparison row*. For every *comparison row*, the algorithm was supposed to detect and mark the standard sample that was most similar to the fired cartridge sample. The 50_75 model (Section Model-based analysis of standard samples) method was applied to a comparison process. However, the classification power of the model failed and was successful in less than 50% of the classification attempts. The main reason might be the fact that the model was built on standard scent sample rather than on scent residue samples. This probably led to a selection of peak area rations significant for distinguishing between standard scent samples, not shot cartridge ones. Model 75_50 (Section Model-based analysis of fired cartridge samples) performed with the same results (successful in less than 50% of the classification attempts) for Pearson's correlation, cosine similarity, and Kendall's tau. However, the Spearman's correlation reached a 100% success rate (see Fig 8) because it was built on the

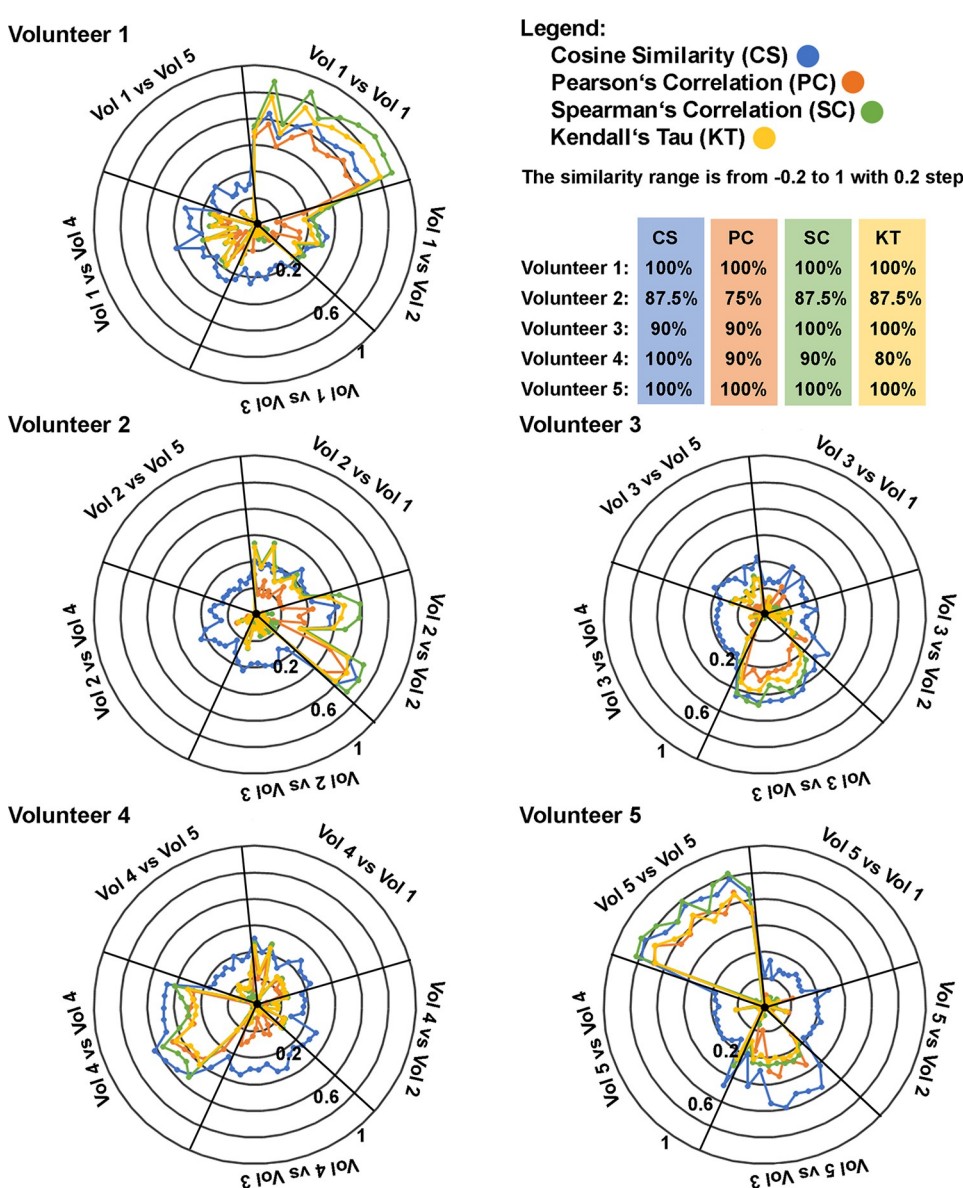

**Fig 7. Graphical representations of similarity results between fired cartridge samples (10 in total) from each volunteer (Vol) against all other standard samples (47 in total).** Radial black lines split the circle into five areas corresponding to the sample sets for each volunteer. The closer the color line is to the edge of the circle, the more similar the compared samples are. For each volunteer, the percentage number corresponds to the percentage of standard samples from the volunteer that reached the top correlation values when compared to their sample collected from fired cartridge. Percentage values below 100% mean that there were standard scent samples from other volunteers which were more similar to the fired cartridge sample than the standard samples from the scent originator.

significant peak area ratios detected on the shot cartridges. However, the standard scent sample of Vol 4 had a higher similarity with the fired cartridge sample of Vol 6. The relevant ratios are listed in S2 Table.

To inspect if the top two correlations are significantly different, both results were transformed to the *z-scores* and the test for the significance of the difference [36, 37] between them was calculated with *p-value* set to 0.05 (see Table 1).

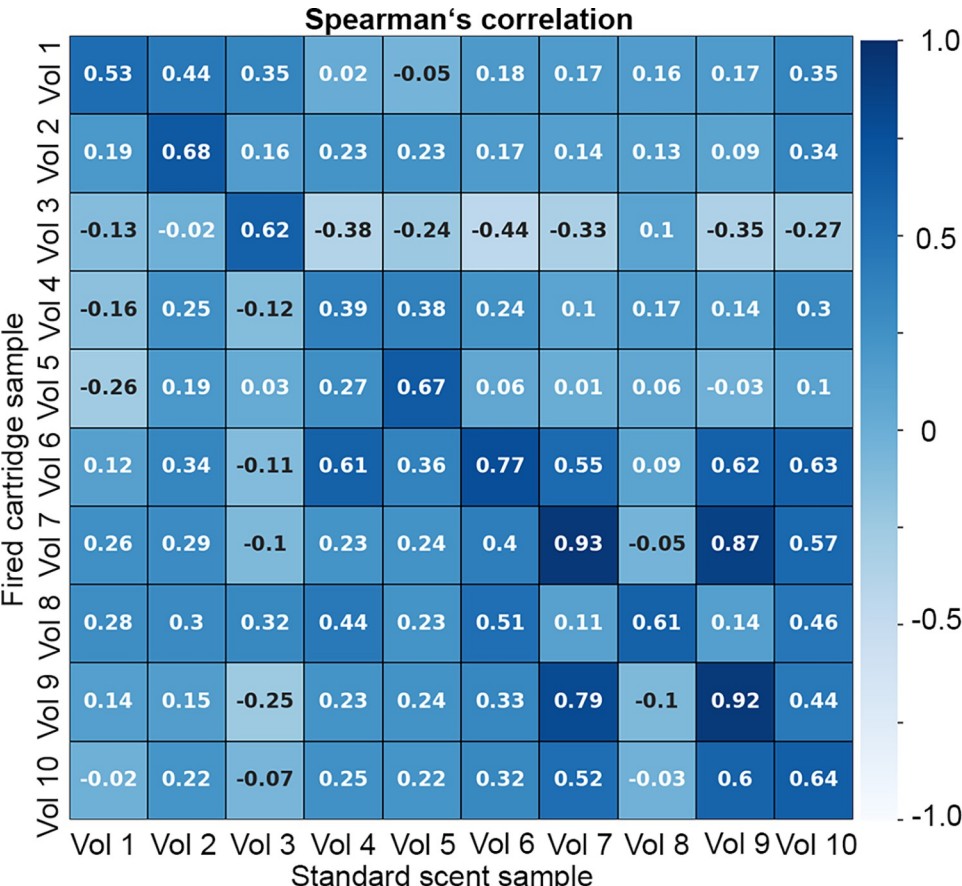

**Fig 8. Graphical representation of the comparison between the standard scent samples and fired cartridge samples.** The values correspond to the Spearman´s correlation score.

## Conclusions

The scent samples collected from the fired cartridges indicated an analogous chemical composition to the scent samples undamaged by shooting collected on the glass beads. The results of the qualitative analysis of scent residues found on the cartridges suggested the sebaceous gland products and their derivatives as the compounds of the main interest for the rest of the conducted experiments. PCA models were developed based on the peak area ratios of the sampled volunteers (Section Model-based analysis of fired cartridge samples). The significant peak area ratios for the comparison identification were selected (Section Data processing) and used for

**Table 1. Test for the significance of the difference between top two Spearman correlation scores for each cartridge sample.**

| Scent sample | Top two scores | p-value | Difference significant? | Scent sample | Top two scores | p-value | Difference significant? |
|---|---|---|---|---|---|---|---|
| Vol1 | 0.5256 0.4402 | 0.5514 | No | Vol6 | 0.7652 0.6280 | 0.1487 | No |
| Vol2 | 0.6800 0.3391 | 0.0110 | Yes | Vol7 | 0.9302 0.8687 | 0.0762 | No |
| Vol3 | 0.6249 0.0972 | 0.0001 | Yes | Vol8 | 0.9221 0.7919 | 0.0050 | Yes |
| Vol4 | 0.3922 0.3798 | 0.9380 | No | Vol9 | 0.6128 0.5063 | 0.4060 | No |
| Vol5 | 0.6696 0.2735 | 0.0047 | Yes | Vol10 | 0.6427 0.5999 | 0.7095 | No |

$H_0$: There is no significant difference between 1st and 2nd top Spearman correlation scores, critical p-value = 0,05.

the identification experiments. To prove the concept of presented workflow, the basic classification algorithms such as K-nearest neighbor and Random Forest were trained on the input data. The most successful model was RF with $F_1$ score equal to 1. As the algorithms were successful in a term of classification, the next step was the similarity comparison. The most suitable similarity comparison was the Spearman´s correlation (Section Similarity analysis of fired cartridge samples versus standard scent samples). The final comparison mimicking the sample comparison performed by the specially trained dogs (Section Fired cartridge sample versus standard scent sample comparison) was 100% successful. If the experiment considered standard scent samples as unknown samples, the method would make one error, misclassifying Volunteer 4 as Volunteer 6. The difference between two top correlations was statistically significant in four out of ten cases.

The objective of this study was to point out the forensic potential of scent evidence found on the fired cartridges for the scent identification. It is important to point out that all steps of the study were conducted under laboratory conditions that did not involve real-world contaminations such as soil contamination, scents of different origin, or any other environmental contaminations; every step of this experiment was designed to reduce the risk of contamination as much as possible, and thus answer the question if there is a possibility to discover patterns in the residues of human scent after being exposed to gunshot. The biggest limitation of this kind of study so far is the available instrumentation. Even with an advanced approach such as GC×GC, the number of chemical compounds below the level of detection might be in the tens or hundreds. With improved sensitivity, the additional higher esters of the fatty acids could be quantified and involved in similarity experiments, which could lead to more convincing results.

## Supporting information

**S1 File. Detailed results of the KNN and RF approaches with links to developed scripts and mathematical formulas used for the calculations.**
(PDF)

**S2 File. Official wording of the written consent collected from participating volunteers.**
(PDF)

**S3 File. Original data used as input for the study along with data results (see file S3 File).**
(RAR)

**S1 Fig. Visual comparison of chromatograms.** A = area of saturated and unsaturated hydrocarbons (the cluster most likely originated from gun lubricants); B = Squalen; C = Esters of higher fatty acids; D = Cholesterol; E = Ethyl esters of carboxylic acids; F = Heterocycles, amides, and more polar compounds in general.
(TIF)

**S1 Table. Summary of the most abundant chemical compounds found in samples (26) collected from fired cartridges.**
(XLSX)

**S2 Table. List of relevant ratios for Spearman's correlation (The model 75_50; Section 3.3.2).**
(XLSX)

**S3 Table. Overall results of similarity analyses of fired cartridge samples versus standard scent samples.**
(XLSX)

**S1 Graphical abstract.**
(TIF)

## Author Contributions

**Conceptualization:** Nikola Ladislavová.

**Funding acquisition:** Štěpán Urban.

**Investigation:** Nikola Ladislavová, Pavel Vrbka.

**Methodology:** Nikola Ladislavová.

**Resources:** Pavel Vrbka.

**Software:** Nikola Ladislavová.

**Supervision:** Štěpán Urban.

**Visualization:** Nikola Ladislavová.

**Writing – original draft:** Nikola Ladislavová.

**Writing – review & editing:** Petra Pojmanová, Jana Šnupárková, Štěpán Urban.

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
