## [Decision Letter · Decision Letter 0]

6 Nov 2022

PONE-D-22-24020Human Scent Signature on Cartridge Case Survives Gun Being Fired: A Preliminary Study on a Potential of Scent residues as an Identification ToolPLOS ONE

Dear Dr. Ladislavova,

Thank you for submitting your manuscript to PLOS ONE. After careful consideration, we feel that it has merit but does not fully meet PLOS ONE’s publication criteria as it currently stands. Therefore, we invite you to submit a revised version of the manuscript that addresses the points raised during the review process.

Please be sure to make all supporting data available. The differences between the laboratory conditions of the reported experiments verses realistic forensic conditions should be discussed. 

We look forward to receiving your revised manuscript.

Kind regards,

James R. Lyons, PhD

Academic Editor

PLOS ONE

Journal Requirements:

2. Please note that PLOS ONE has specific guidelines on code sharing for submissions in which author-generated code underpins the findings in the manuscript. In these cases, all author-generated code must be made available without restrictions upon publication of the work. Please review our guidelines at https://journals.plos.org/plosone/s/materials-and-software-sharing#loc-sharing-code and ensure that your code is shared in a way that follows best practice and facilitates reproducibility and reuse

Reviewers' comments:

Reviewer's Responses to Questions

**Comments to the Author**

1. Is the manuscript technically sound, and do the data support the conclusions?

Reviewer #1: Partly

2. Has the statistical analysis been performed appropriately and rigorously? 

Reviewer #1: No

3. Have the authors made all data underlying the findings in their manuscript fully available?

Reviewer #1: No

4. Is the manuscript presented in an intelligible fashion and written in standard English?

Reviewer #1: No

5. Review Comments to the Author

Reviewer #1: The submitted manuscript reports a study on the survival of human scent on fired gun cartridges, and on the possibility to identify the shooter based on the GCxGC-MS analysis of scent residues samples and that of standard samples obtained under laboratory conditions using a similarity analysis of an adequate scent representation. The authors claim that though contaminated by the gun, the scent survives to a great extent and enables individual identification.

General comments

1) The conclusion should acknowledge that the scent survival on the cartridges is obtained after a procedure that is closer to laboratory conditions than to a realistic forensic context, namely 5 minutes of hand rubbing to activate the sebaceous glands of the subjects and 10 minutes of rubbing the cartridges.

2) The choice of a scent representation as a selection of peak area ratios is perfectly justified (among others by Curran’s studies and results), as well as choosing them according to their rate of occurrence and stability across samples. However, it is not clearly described how the number of ratios is selected using cross-validation on “PCA models”, nor what exactly is meant by a “PCA model” (e.g. a model defined by a number of principal components). Without further information, the selection procedure is not replicable.

3) Two scent representations are established, a first one using the scent residue samples, and a second one using the standard samples. It is not clear which of the two was used to match the fired cartridge samples with the standard ones using the similarity analyses on the set of subjects used to establish these representations. Also, why not test both representations, and also consider a representation based on both standard and scent residue samples.

4) The fact that, for the final comparison on a new set of subjects, the representation based on standard samples does not at all enable to match the fired cartridge samples with the standard ones whereas the representation based on scent residues does when using Spearman’s similarity measure would be better interpretable if a) the same comparison had been made on the subjects used to establish the representations, and b) both representations were given (the identity of the ratios is provided in the supporting information only the representation based on the scent residues).

5) The absence of section numbering and many errors in the references to the figures make the manuscript very difficult to read.

6) The manuscript would benefit from being proofread by a native English speaker.

Specific comments

• p. 6, line 120, “all n detected peaks”. What is the value of n?

• p. 6, line 129, “the N number of peak area ratios with the least standard deviation (sorted in ascending order) through all samples were included in further calculations”. 1) Is it an intrasubject standard deviation? 2) How is N determined?

• p. 6, line 131, “the incidence rate threshold was established”. 1) How ? 2) why is a second incidence rate threshold needed (the first one being 75% for each subject)?

• p. 6, line 134, “in the presented PCA models”. What PCA models? Figures 3 and 5 display score plots on the two first principal components, and the captions the percentage of the variance explained by the first 7 principal components. What is relevant to the selection?

• p. 7, line 138: why is normality tested for?

• p. 7, line 140: “PCA [14] was applied as a model-based method”. If the authors mean that they evaluate the performance as the percentage of variance explained by the m first components, this should be stated (with the value of m). The selection of N and of the incidence rate threshold should be clarified here.

• p. 9, line 198: Figure 3 instead of 2.

• p. 11, line 230: Figure 5 instead of 4.

• p. 11, Fig. 5 caption: “Model with 50 ratios with least SD and with equal or less that 75% zero value for each ratio through the standard and fired samples (57 ratios in total) using leave-one out validation”: 1) how can 50 selected ratios become 57? 2) Since this model is based on the standard samples, why is the occurrence rate through standard AND fired samples?

• p. 11, section “Similarity analysis of fired cartridge samples versus standard scent samples”: which one of the two models is used, that based on the scent residue samples, or that based on the standard samples? The text does not tell, nor does the diagram of Figure 2.

• p. 13, line 276: “The model 75_50 () performed with the same results for Pearson’s correlation, cosine similarity and Kendall’s tau.” What results? How to they compare to Spearman’s? Especially the cosine similarity whose performance is very close to Spearman’s according to Table E.1.

• p. 13, line 278, “The Spearman’s correlation reached a 100% success rate”: it is true, when looking what standard scent sample best matches a given fired cartridge sample, but not conversely (vol. 4 is matched with vol. 6).

6. PLOS authors have the option to publish the peer review history of their article (what does this mean?). If published, this will include your full peer review and any attached files.

Reviewer #1: No

---

## [Author Response · Author response to Decision Letter 0]

19 Dec 2022

Journal Requirements:

1. We reworked problematic areas of the manuscript

2. All coding used for the evaluation of the result was stored at GitHub and properly licenced.

3. Written consent was added

Response to Reviewers:

Manuscript: Human Scent Signature on Cartridge Case Survives Gun Being Fired: A Preliminary Study on a Potential of Scent residues as an Identification Tool (PLOS ONE) - the original file is attached to hte submission.

• p. 6, line 120, “all n detected peaks”. What is the value of n?

The argument 'all n detected peaks' was rewritten to 'all 75 selected peaks'.

• p. 6, line 129, “the N number of peak area ratios with the least standard deviation (sorted in ascending order) through all samples were included in further calculations”. 1) Is it an intrasubject standard deviation? 2) How is N determined?

1) Yes, is the intrasubject SD

2) N is a parameter chosen by the scientist responsible for the experiment. The value can be set as an integer (rigid count of SDs from each subset will be chosen), or SD value threshold can be set (then the number of the SDs will most likely vary among the subsets). We have chosen the first approach, because we found the second approach more viable for normally distributed dataset. The comment was added to the body of the article [lines 157- 160]

• p. 6, line 131, “the incidence rate threshold was established”. 1) How ? 2) why is a second incidence rate threshold needed (the first one being 75% for each subject)?

1) This parameter is also chosen by the scientist responsible for the experiment. The lower tolerance to zero values means that only the common chemical compounds are kept for further analysis. The higher tolerance to zero values, on the other hand, involves even the subject-specific chemical compounds in the following analysis [lines 148 – 150].

2) The second incidence rate threshold was established because we wanted to eliminate subject-specific chemical compounds and keep only chemical compounds common for all (or majority of) subjects.

• • p. 6, line 134, “in the presented PCA models”. What PCA models? Figures 3 and 5 display score plots on the two first principal components, and the captions the percentage of the variance explained by the first 7 principal components. What is relevant to the selection?

Only the first two components were demonstrated, because they describe most of the variance. The rest of the percentage of described variance is just for informational reasons. Recalculation of the PCA models was done with a code provided to make it accessible for every reader. Another file (Calculations) was added to Supporting Information and described the models in more depth with the code provided.

• • p. 7, line 138: why is normality tested for?

It is another piece of information that we used to support our results. The finding that our data are not normal corresponds to a higher success rate of Spearman’s and Kindal’s correlations. It was very helpful when deciding how to choose N (mentioned in the previous question)

• • p. 7, line 140: “PCA [14] was applied as a model-based method”. If the authors mean that they evaluate the performance as the percentage of variance explained by the m first components, this should be stated (with the value of m). The selection of N and of the incidence rate threshold should be clarified here.

The statements were clarified in the body of the text (lines 162–166)

• p. 9, line 198: Figure 3 instead of 2.

Fixed numbering.

• p. 11, line 230: Figure 5 instead of 4.

Fixed numbering.

• p. 11, Fig. 5 caption: “Model with 50 ratios with least SD and with equal or less that 75% zero value for each ratio through the standard and fired samples (57 ratios in total) using leave-one out validation”: 1) how can 50 selected ratios become 57? 2) Since this model is based on the standard samples, why is the occurrence rate through standard AND fired samples?

1) 50 ratios from each volunteer (5 volunteers in total). The final set contained 57 unique ratios. Theoretically, if all volunteers had unique ratios that were not shared with the rest, we would end up with 250 unique ratios in our model.

2) We took every possible step to ensure that the relevant ratios for the final similarity comparison are present in both the standard and the scent samples.

• p. 11, section “Similarity analysis of fired cartridge samples versus standard scent samples”: which one of the two models is used, that based on the scent residue samples, or that based on the standard samples? The text does not tell, nor does the diagram of Figure 2.

Both ‘Similarity analyses’ were transformed into subsections of the corresponding model sections to improve readability.

• p. 13, line 276: “The model 75_50 () performed with the same results for Pearson’s correlation, cosine similarity and Kendall’s tau.” What results? How to they compare to Spearman’s? Especially the cosine similarity whose performance is very close to Spearman’s according to Table E.1.

Yes, if we look at the average values, the CS is doing as well as SC. However, if we look closely at Fig 6, the absolute difference between the average correlation value of class 1 (samples from the same volunteer) and class 0 (samples from different volunteers) was greater for Spearman’s than for cosine.

• p. 13, line 278, “The Spearman’s correlation reached a 100% success rate”: it is true, when looking what standard scent sample best matches a given fired cartridge sample, but not conversely (vol. 4 is matched with vol. 6).

The reviewer is right; we added this statement into the corresponding section and Conclusion.

---

## [Decision Letter · Decision Letter 1]

22 Dec 2022

PONE-D-22-24020R1Human Scent Signature on Cartridge Case Survives Gun Being Fired: A Preliminary Study on a Potential of Scent residues as an Identification ToolPLOS ONE

Dear Dr. Ladislavova,

Thank you for submitting your manuscript to PLOS ONE. After careful consideration, we feel that it has merit but does not fully meet PLOS ONE’s publication criteria as it currently stands. Therefore, we invite you to submit a revised version of the manuscript that addresses the points raised during the review process.

We look forward to receiving your revised manuscript.

Kind regards,

James R. Lyons, PhD

Academic Editor

PLOS ONE

Additional Editor Comments:

Please reply completely and in detail to the reviewer's requests. The reviewer believes that many of their requests were not addressed in the first revision.

Reviewers' comments:

Reviewer's Responses to Questions

**Comments to the Author**

1. If the authors have adequately addressed your comments raised in a previous round of review and you feel that this manuscript is now acceptable for publication, you may indicate that here to bypass the “Comments to the Author” section, enter your conflict of interest statement in the “Confidential to Editor” section, and submit your "Accept" recommendation.

Reviewer #1: (No Response)

2. Is the manuscript technically sound, and do the data support the conclusions?

Reviewer #1: Partly

3. Has the statistical analysis been performed appropriately and rigorously? 

Reviewer #1: No

4. Have the authors made all data underlying the findings in their manuscript fully available?

Reviewer #1: No

5. Is the manuscript presented in an intelligible fashion and written in standard English?

Reviewer #1: No

6. Review Comments to the Author

Reviewer #1: In their response to reviewers, the authors do not answer the reviewer’s main comments. There is still no section numbering. The notations for the parameters and models are still unclear.

The answers to the detailed comments are often in contradistinction with the corrections made in the revised manuscript, and/or not self-contained or still not precise enough. See the answers to my first two comments for example:

• p. 6, line 120, “all n detected peaks”. What is the value of n?

Answer in the response to the reviewers : “The argument 'all n detected peaks' was rewritten to 'all 75 selected peaks'.”

Answer in the revised manuscript : n vanished, we read on p. 6 lines 132-134: “After data alignment via RIs, all detected peaks were inspected, and the subset of 101 selected chemical compounds presented in standard and fired sample datasets was created and further processed.” 'all 75 selected peaks' appears nowhere.

• p. 6, line 129, “the N number of peak area ratios with the least standard deviation (sorted in ascending order) through all samples were included in further calculations”. 1) Is it an intrasubject standard deviation? 2) How is N determined?

Answer in the response to the reviewers :

1) Yes, is the intrasubject SD

2) N is a parameter chosen by the scientist responsible for the experiment. The value can be set as an integer (rigid count of SDs from each subset will be chosen), or SD value threshold can be set (then the number of the SDs will most likely vary among the subsets). We have chosen the first approach, because we found the second approach more viable for normally distributed dataset. The comment was added to the body of the article [lines 157- 160]

Lines 157 to 160 the revised manuscript:

Information about the (non)normality of the distribution was important for the correct estimation of the parameter N value and an evaluation of the results from the similarity

standard deviation threshold value, but this approach was not suitable for non-normal distributed data. Therefore, an approach of setting the rigid count of the peak area ratios with the least standard deviation was prioritized. Two different statistical approaches were used for the following data analyzes: Principal Component Analysis and similarity comparison method. Principal Component. Analysis (PCA) [14] was used solely as a visualization of possible trends in data variance and as a tool for parameter optimization – models with higher percentage of variance explained for 7 principal components were considered as more reliable. Different sets of zero-value filters (incidence rate threshold) and number of characteristic ratios (N – see section Data processing) were tested.

So is the percentage of variance explained by the with 7 PC of a PCA model used to select the value of N? If yes, why 7? What configurations of N and of the two incidence rate ratios were tested?

Furthermore, some corrections/additions in the revised manuscript raise more questions than they provide clarifications. For example:

• p. 7, lines 147-148: “This threshold was also estimated and optimized.” This isolated sentence is added without any reference to the section where the optimization takes place.

Etc.

This is not a proper revision.

7. PLOS authors have the option to publish the peer review history of their article (what does this mean?). If published, this will include your full peer review and any attached files.

Reviewer #1: No

---

## [Author Response · Author response to Decision Letter 1]

4 Feb 2023

Reviewer's Responses to Questions:

4. Have the authors made all data underlying the findings in their manuscript fully available?

Reviewer #1: No

We added the summary data (which were the starting point of our calculations) to a submission. All scripts used in the manuscript were already published in an open repository via Github. 

5. Is the manuscript presented in an intelligible fashion and written in standard English?

Reviewer #1: No

The typos were checked via software, and we consulted the text with a native speaker. Can the Reviewer elaborate a bit on this issue?

6. Review Comments to the Author:

There is still no section numbering.

We even checked already published articles. They have no section numbering. The guidelines, and the sample manuscript body do not mention section numbering, sample manuscript has no numbering.

The notations for the parameters and models are still unclear.

We changed the notations of incidence thresholds to fix the problem with the notations. We have rewritten the incidence threshold to be corresponding with the used script as it was misleading. In the script, the user input is set to:

zerocount = int(input("Mow many zeros in column are acceptable? [%] "))

The incidence threshold should be referring to zero values. This issue is addressed in the body of the text.

Models are already referred in the text with the section reference as well as with the code. We added the naming policy to the body of the text [lines 152-153], hopefully it will help with the navigation. Both similarity comparison sections are already a subsection of the Model sections, so it is clear which model is used for the data pre-processing (because the PCA is not an identification tool, just a visualization).

• p. 6, line 120, “all n detected peaks”. What is the value of n?

Answer in the response to the reviewers : “The argument 'all n detected peaks' was rewritten to 'all 75 selected peaks'.”

Answer in the revised manuscript : n vanished, we read on p. 6 lines 132-134: “After data alignment via RIs, all detected peaks were inspected, and the subset of 101 75 selected chemical compounds presented in standard and fired sample datasets was created and further processed.” 'all 75 selected peaks' appears nowhere.

Reviewer is right, the original subset involved 101 peaks. The author was referring to a subset of 75 peaks in the Responses (after we excluded peaks appearing only in few samples). The Author is sorry for the confusion. ‘75 peaks’ is now in the text. The original summary table is split into three sections according to the samples and attached to the manuscript to improve point 5. Please note that the table containing only cartridge samples contains 59 peaks as 16 peaks had zero values in all cartridge samples.

So is the percentage of variance explained by the with 7 PC of a PCA model used to select the value of N? If yes, why 7? What configurations of N and of the two incidence rate ratios were tested?

No, the percentage of variance was not as significant as the distance and separation of each observed cluster on the final plot. We clearly stated that PCA was solely explorative tool. The higher the total explained variance, the better of course. However, the distance of the Volunteers clusters was the crucial parameter. First incidence rate was always set to 75% [line 141], the second one was set in range from 25 to 75 with step of 25. The N parameter was tested in range from 25 up to 500 with step of 25. We added the ranges into the body of the text.

Furthermore, some corrections/additions in the revised manuscript raise more questions than they provide clarifications. For example:

• p. 7, lines 147-148: “This threshold was also estimated and optimized.” This isolated sentence is added without any reference to the section where the optimization takes place.

This refers to the second incidence rate threshold.

Etc.

This is not a proper revision.

We decided to do an additional statistical analysis via KNN and RN to support our concept, because we feel from the Reviewer’s comments that explorative PCA as proof of the concept is not enough. For the final comparison, we added the test for significant differences for the top two correlation coefficients in the final sample comparison.

We hope this will improve the soundness of the results from the statistical point of view.

---

## [Editor Report · Decision Letter 2]

6 Mar 2023

Human Scent Signature on Cartridge Case Survives Gun Being Fired: A Preliminary Study on a Potential of Scent residues as an Identification Tool

PONE-D-22-24020R2

Dear Dr. Ladislavova,

We’re pleased to inform you that your manuscript has been judged scientifically suitable for publication and will be formally accepted for publication once it meets all outstanding technical requirements.

Kind regards,

James R. Lyons, PhD

Academic Editor

PLOS ONE
---

## [Editor Report · Acceptance letter]

13 Mar 2023

PONE-D-22-24020R2 

Human Scent Signature on Cartridge Case Survives Gun Being Fired: A Preliminary Study on a Potential of Scent Residues as an Identification Tool 

Dear Dr. Ladislavova:

I'm pleased to inform you that your manuscript has been deemed suitable for publication in PLOS ONE. Congratulations! Your manuscript is now with our production department. 

Kind regards, 

on behalf of

Dr. James R. Lyons 

Academic Editor

PLOS ONE